# Prebiotic and Anti-Adipogenic Effects of Radish Green Polysaccharide

**DOI:** 10.3390/microorganisms11071862

**Published:** 2023-07-24

**Authors:** Yu Ra Lee, Hye-Bin Lee, Yoonsook Kim, Kwang-Soon Shin, Ho-Young Park

**Affiliations:** 1Food Functionality Research Division, Korea Food Research Institute, Wanju-gun 55365, Republic of Korea; lyr@kfri.re.kr (Y.R.L.); l.hyebin@kfri.re.kr (H.-B.L.); kimyus@kfri.re.kr (Y.K.); 2Department of Food Science and Biotechnology, Kyonggi University, Suwon 16227, Republic of Korea; ksshin@kyonggi.ac.kr; 3Department of Food Biotechnology, University of Science and Technology, Daejeon 34113, Republic of Korea

**Keywords:** radish greens, *Raphanus sativus*, polysaccharide, prebiotic, short-chain fatty acid, anti-adipogenic

## Abstract

Radish (*Raphanus sativus* L.) greens are consumed as a source of nutrition, and their polysaccharides such as rhamnogalacturonan-I possess certain beneficial properties. This study investigated the prebiotic effects of a radish green polysaccharide (RGP) on gut health and obesity. The prebiotic activity of RGP was evaluated based on the pH changes and short-chain fatty acids (SCFAs) concentration. The results showed that 0.5% RGP had a higher prebiotic activity score than inulin and increased SCFAs production in all five prebiotic strains. Moreover, RGP inhibited fat accumulation in 3T3-L1 adipocytes, indicating its potential to reduce obesity. Overall, these findings suggested that the polysaccharide of radish greens has prebiotic effects and may serve as a beneficial prebiotic for gut health and obesity.

## 1. Introduction

Radish (*Raphanus sativus* L.) is a common vegetable that belongs to the Brassicaceae family. It has various health benefits, such as anti-cancer, anti-diabetic, anti-hypertensive, and anti-obesity effects [1]. Radish greens are the edible leaves of radish plants, which are often discarded or used as animal feed. However, radish greens are rich in nutrients and bioactive compounds, such as polysaccharides [2]. Radish greens, the upper part of the radish root, including stems and leaves, are a popular delicacy among Koreans. However, their limited harvest season and lack of commercialization have hindered them from being traditional specialty foods. A previous study showed that a water-soluble radish green extract reduces body fat and improves the gut environment in mice [3].

The development of obesity in adults is closely associated with adipogenesis, which involves the differentiation of preadipocytes into adipocytes. The process assumes significance as obese individuals tend to have a higher number of fat cells compared to their lean counterparts [4]. Recent advancements in the field of food and nutrition sciences have brought attention to the potential of modulating specific physiological functions within the body through dietary intake [5]. Especially, functional foods targeting obesity may encompass various bioactive components such as fatty acids, phenolic compounds, soybean products, plant sterols, calcium-rich foods, and dietary fiber [6]. Research on the anti-adipogenic effects of various foods is being conducted, and specifically, studies have indicated the potential of the ginsenoside Rg3 [7], chitosan oligomers [8], and water-soluble polysaccharide from *Lycium barbarum* L. [9] as a promising factor in preventing obesity by inhibiting the differentiation of human primary adipocyte precursors.

Plant polysaccharides are complex carbohydrates that are not digested by the host but can be used by beneficial bacteria in the gut [10]. These bacteria produce short-chain fatty acids (SCFAs) and other metabolites that improve the host’s health [11]. Some examples of plant polysaccharides that act as prebiotics are plant cell wall polysaccharides, such as cellulose, hemicellulose, and pectin, and plant storage polysaccharides, such as starch and fructans [12]. Also, polysaccharides, a type of dietary fiber, play a crucial role in preventing various diseases. For instance, pectins, inulin, and gums have the ability to slow down the digestion process [13], lower blood cholesterol levels [14], and regulate the absorption of sugars into the bloodstream, preventing sudden spikes in blood sugar levels [15]. Pectin is a type of pectic polysaccharide that is found in many fruits and vegetables, such as apples, oranges, and radish greens. Pectin has various effects on the gut microbiota, such as increasing the abundance of *bifidobacteria* and *lactobacilli*, modulating the immune system and enhancing the intestinal barrier function [16]. Plant polysaccharides are also important for biotechnology, as they can be converted into biofuels, bioplastics and novel functional foods.

The gut microbiota comprises diverse and complex communities, including bacteria, fungi, and protozoa. Such microbiota plays a crucial role in promoting the overall health of the host through metabolic activities and physiological regulation, such as nutrient absorption, bioactive compound synthesis, improving intestinal barrier function and motility, resistance against pathogens, and modulating the immune system [17,18]. These functions significantly affect the overall health of the host, establishing the gut microbiota as a key component of the body. Prebiotics are low-molecular-weight fibers that evade breakdown by digestive enzymes in the stomach and small intestine, and help improve the gut environment [19]. Inulin, a naturally occurring oligosaccharide stored as a carbohydrate reserve in thousands of plants [20], is the most well-known prebiotic compound and is often used as a positive control to confirm prebiotic activity. Inulin is a fiber that is easily fermented by gut bacteria, resulting in the production of large amounts of SCFAs [21]. Hence, comparison with inulin is essential in research on human gut microbial communities. SCFAs are important metabolites that maintain intestinal homeostasis [22]. They commonly lower the intestinal pH to maintain an acidic environment, prevent the colonization of harmful bacteria, and aid in nutrient absorption.

Previous studies have shown that polysaccharide fraction from radish greens has anti-obesity effects in high-fat-diet-induced obese mice by improving gut barrier function and regulating lipid metabolism [2]. However, the prebiotic and anti-adipogenic effects of radish greens polysaccharide (RGP) have not been fully elucidated. Therefore, the aim of this study was to investigate the beneficial effects of RGP on probiotic bacteria and adipogenesis in vitro. We hypothesized that RGP could modulate the growth rate of beneficial microbes and inhibit the differentiation and lipid accumulation of adipocytes. Our findings have important implications for developing novel food products and supplements to improve gut health.

## 2. Materials and Methods

### 2.1. Preparation of RGP

RGP was prepared using the methods described by previous study [2]. In brief, edible radish greens (*R. sativus* L.) cultivated in Wanju-gun, Republic of Korea were dried under 40 °C and grounded radish greens powder (100 g) was extracted in 2 L of distilled water for 3 h at 80 °C. After filtration and concentration using a vacuum rotary evaporator (R-114; Buchi Labortechnik, Flawil, Switzerland), the concentrates were precipitated with four volumes of ethanol for 16 h at 4 °C. To obtain a crude polysaccharide precipitate, the alcohol precipitation solution is subjected to centrifugation using a high-speed centrifuge. The resulting precipitate is washed twice with a small quantity of absolute ethanol. This process yields a crude polysaccharide product with an extraction rate of 4.9%.

### 2.2. Composition Analysis of RGP

This study analyzed the composition of RGP using various analytical methods. The measurement of neutral sugar content was conducted utilizing the phenol-sulfuric acid method [23], whereas uronic acid content was analyzed using the *m*-hydroxybiphenyl method [24]. Total polyphenol content was assessed using the Folin–Ciocalteu spectrophotometric method [25], and the protein content was analyzed using the Bradford assay (Bio-Rad, Hercules, CA, USA). The KDO content was determined using the thiobarbituric acid method [26], and the monosaccharide composition using the alditol acetate method [27] with slight modifications.

The sugar composition of RGP was analyzed using gas chromatography (GC) with a modified alditol acetate method. A Young-Lin Co. ACME-6100 GC instrument, equipped with a SP-2380 capillary column (Supelco, Bellefonte, PA, USA) and a flame ionization detector (FID), was employed for the analysis. The GC temperature program included the following steps: 60 °C (1 min), 60 → 220 °C (30 °C/min), 220 °C (12 min), 220 → 250 °C (8 °C/min), and 250 °C (15 min). Molecular ratios were determined by calculating peak areas and applying appropriate response factors.

### 2.3. Prebiotic Activity Assay

The prebiotic effects of the samples were determined as a prebiotic activity score using the modified method of Lee et al. [28]. In brief, five probiotic strains were used in this study: *B. bifidum* (MG731, infant gut-origin), *B. longum* (MG723, infant gut-origin), and *Lacticaseibacillus paracasei* (information), *Lactobacillus plantarum* (information), which were kindly provided by Mediogen (Jecheon, Republic of Korea); *Escherichia coli* (KCTC2441), which was purchased from the Korean Collection for Type Cultures (Jeongeup, Republic of Korea); and *Lactobacillus lactis* KF140, which was isolated from fermented kimchi at the Korea Food Research Institute. *Lactobacillus* strains were cultured on Lactobacilli MRS agar and broth. And *E. coli* was cultured on tryptic soy agar and broth at 37 °C for 20–24 h. The *Bifidobacterium* strains were activated in blood and liver (BL) agar and broth anaerobically at 37 °C for 48 h. All broth and agar were purchased from BD DifcoTM (Franklin Lakes, NJ, USA). After culturing in broth, the bacterial strains were centrifuged at 1000× *g* for 5 min and suspended in M9 broth supplemented with 0.1% glucose, 0.0015% CaCl_2_, and 0.05% MgSO_4_. Optical density of 2.5 ± 0.01 (at 600 nm) was determined using a spectrophotometer (Jenway 6715, Cole-Parmer Ltd., Vernon Hills, IL, USA).

The prebiotic activity score presents the ability of the substrate to exhibit the growth of probiotic strains and not those of enteric bacteria (*E. coli*). Diluted bacterial cultures were inoculated at a concentration of 1% (*v*/*v*) into respective tubes containing M9 medium supplemented with prebiotics (inulin, RGP) or glucose. The bacterial cultures, adjusted for absorbance, were incubated at 37 °C for 24 h in the presence of 0.5% (*w*/*v*) glucose, 0.5% (*w*/*v*) inulin, or 0.1% and 0.5% (*w*/*v*) RGP dissolved in M9 broth. The prebiotic activity score was determined using the provided equation:
$$\begin{array}{l} \mathrm{Prebiotic}\;\mathrm{activity}\;\mathrm{score}\\ \;\;\;\;\;\;\;\;=\frac{probiotic\;log\;CFU/mL\;on\;the\;prebiotic\;at\;24\;h\;-\;probiotic\;log\;CFU/mL\;on\;the\;prebiotic\;at\;0\;h}{probiotic\;log\;CFU/mL\;on\;glucose\;at\;24\;h\;-\;probiotic\;log\;CFU/mL\;on\;glucose\;at\;0\;h}\\ \;\;\;\;\;\;\;\;-\frac{enteric\;log\;CFU/mL\;on\;the\;prebiotic\;at\;24\;h\;-\;enteric\;log\;CFU/mL\;on\;the\;prebiotic\;at\;0\;h}{enteric\;log\;CFU/mL\;on\;glucose\;at\;24\;h\;-\;enteric\;log\;CFU/mL\;on\;glucose\;at\;24\;h} \end{array}$$

### 2.4. Total SCFAs Analysis Using GC-Flame Ionization Detector (FID)

GC analysis was performed using an ACME 6100 (Young-Lin Co., Ltd., Anyang, Republic of Korea) equipped with a capillary column (SP-2380, 0.25 mm × 30 m, 0.2 μm-film thickness; Supelco, Bellefonte, PA, USA). The optimal temperature conditions [60 °C (1 min), 60 °C → 180 °C (30 °C/min), 180 °C → 250 °C (1.5 °C/min), 250 °C (5 min)] were employed for the analysis in splitless injection mode (1/20). The carrier gas (N_2_) flow rate was adjusted to 1.5 mL/min. The molar percentage was determined by calculating the ratios of peak areas and applying response factors, utilizing an FID [27]. The total SCFAs were analyzed as the sum of butyric acid, acetic acid, and propionic acid.

### 2.5. Cell Culture and Viability Assay

Mouse 3T3-L1 preadipocytes were purchased from the American Type Culture Collection (ATCC, Manassas, VA, USA) and cultured in Dulbecco’s modified Eagle’s medium (DMEM; HyClone Laboratories, Helsinki, Finland) containing 10% bovine calf serum (Gibco/BRL, Gaithersburg, MD, USA), 100 U/mL penicillin, and 10 mg/mL streptomycin at 37 °C and 5% CO_2_. Cell viability was analyzed using the Cell Counting Kit-8 (CCK-8; Dojindo, Tokyo, Japan) following the manufacturer’s instructions.

### 2.6. Determination of Lipid Accumulation

Oil Red O (ORO) staining helped determine lipid accumulation. The cells were fixed with 10% formalin for 10 min and then stained with ORO solution for 30 min to visualize the lipid droplets. ORO-stained lipid droplets were observed under an inverted microscope (Eclipse Ti-U; Nikon, Tokyo, Japan). The dye was extracted using isopropanol, and the amount of lipid accumulation was measured by determining the absorbance at 520 nm. A commercial *Garcinia gummi-gutta* extract used as a positive control.

### 2.7. Statistical Analysis

Experimental data are presented as the mean ± standard deviation. All experiments were conducted using a triple-repetition experimental design. Data were analyzed using one-way analysis of variance with Duncan’s multiple range test using SPSS version 20 (SPSS Inc., Armonk, NY, USA). A *p*-value of <0.05 was considered statistically significant.

## 3. Results

### 3.1. Composition Analysis of RGP

As shown in Table 1, RGP was primarily composed of neutral sugars (68.1%), and uronic acids (20.4%), with minor protein components (1.4%). In addition, polyphenols and KDO-like materials contents were 6.5% and 2.2%, respectively. The sugar composition of RGP was galactose (42.1%), arabinose (24.2%), and galacturonic acid (14.0%). Trace amounts of glucuronic acid (2.1%) and xylose (0.4%) were detected. All data are presented as mean ± standard deviation (n = 3).

### 3.2. Prebiotic Effects of RGP

A prerequisite for a food to be categorized as a prebiotic is its ability to selectively modulate the growth of gut microbiota, specifically promoting the growth of probiotic strains like *Lactobacillus* and *Bifidobacterium* [2,29]. The gut microbiota is closely associated with an individual’s health, and probiotic strains residing in the gut exert numerous beneficial effects on host physiology, including pathogen inhibition and immune enhancement [30]. The prebiotic effects of RGP were evaluated by determining the prebiotic activity score using five representative probiotic strains. An analysis of the prebiotic activity score after treatment with various bacteria were compared to those of the normal control, positive control (inulin), and different concentrations of RGP (0.1% and 0.5%) (Figure 1A). Inulin is a non-digestible carbohydrate known for selectively stimulating beneficial microbiota and improving gut health [31]. The prebiotic activity scores for *L. plantarum*, *L. lactis*, *B. bifidum*, *B. longum* were 1.61-, 1.45-, 1.89-, and 1.63- fold higher in the HFRL 0.5% group than in the culture treated with inulin. Four probiotic strains have shown the prebiotic activity score to be significantly higher than inulin at the same concentration (0.5%).

We also confirmed the pH of the final medium, as shown in Figure 1B. pH crucially affects prebiotic activity, warranting an appropriate pH range to maintain prebiotic components. Additionally, pH plays an important role in maintaining the balance of gut bacteria. The pH changes over 24 h and it reduces the culture media containing RGP by several bacteria over that time period (Figure 1B). The medium containing RGP at a concentration of 0.5% exhibited comparable or lower prebiotic activity compared to the medium containing inulin. The enhancement in prebiotic activity and the reduction in pH confirmed the efficient utilization of RGP as a carbon source by probiotic strains.

### 3.3. Concentrations of Total SCFAs in Five Probiotic Strains

SCFAs, which are primarily produced through the microbial fermentation of carbohydrates, have been found to possess several beneficial effects. They can inhibit the growth of pathogenic intestinal bacteria and regulate lipid metabolism and the immune system [32]. The total SCFAs levels upon RGP addition were measured (Figure 2), evaluating the effect on SCFAs production. The production of metabolites, including SCFAs, by probiotic strains is a significant contributor to pH reduction [31], and enhance the absorption of essential minerals like magnesium and calcium, thereby inhibiting the proliferation of harmful bacteria [33]. The impact of RGP supplementation on the production of SCFAs by intestinal probiotic bacteria was investigated. The sum of concentrations of acetic, propionic, and butyric acids generated during a 24 h fermentation of RGP by probiotic strains were shown in Figure 2. More SCFAs were produced when RGP was co-cultured with all strains of probiotics. In particular, the RGP 0.5% group medium exhibited significantly higher total SCFAs levels compared to the NC group. Moreover, higher RGP concentrations produced more SCFAs. RGP could produce more SCFAs than inulin in three probiotic strains such as *L. plantarum*, *L. lactis*, and *B. bifidum*, and *B. longum* significantly at the same concentration (0.5%). The specific contents of three SCFAs such as acetic acid, butyric acid, and propionic acid are presented in Appendix A. These findings indicate that cultures utilizing RGP as the carbon source yielded altered SCFAs profiles compared to the NC group for all probiotic strains.

### 3.4. Evaluation of the Lipid-Regulating Capacity of RGP

We exposed 3T3-L1 cells to RGP at varying concentrations (0, 10, 100, 250, 500, and 1000 μg/mL) using cell counting kit 8 assay (Figure 3A) to assess the impact of RGP on adipocyte toxicity. The results showed no significant cellular apoptosis at various concentrations of RGP. Hence, the findings indicated that RGP did not have any impact on the viability of adipocytes at any concentration tested.

The effect of RGP on lipid accumulation was evaluated by ORO staining. As shown in Figure 3B, the lipid accumulation in the RGP100 group was significantly lower than that in the control group.

## 4. Discussion

Recently, more research has been conducted on the potential health benefits of the commonly discarded radish leaves. Radish leaves are believed to have nutritional and medicinal value, particularly fueled by their antioxidant activity. Studies have investigated their potential as functional foods [34]. Given their reported nutritional and medicinal properties, this study explored the effects of radish leaves on human health.

In particular, plant extracts hold promise as potential therapeutics for the prevention and treatment of obesity and related metabolic disorders, with mechanisms involving the gut microbiota [35]. Increasing evidence suggests that obesity and related metabolic disorders are associated with significant alterations in the composition and diversity of the gut microbiota, exerting detrimental effects on the host’s gut microbial structure [36,37]. There has been extensive research on plant extracts, mulberry leaf extracts [38,39], grape seed proanthocyanidin extract [40,41], honokiol [42], capsaicin [43] and konjac glucomannan [44] that can influence the structure of the gut microbiota.

Numerous studies have focused on the inhibitory effects of dietary sugars on obesity. Dietary sugars are known to be composed of various monosaccharides and have various health benefits such as antitumor [45], anti-inflammatory [46], antioxidant, and antidyslipidemic effects [47]. We aimed to investigate the composition of radish extract, which has traditionally been used as a raw material for Kimchi in Korea [48] and exhibits various biological activities, including antimicrobial, antioxidant [49,50], anti-inflammatory [51], and anti-hypertensive ones [52]. Our analysis of immature radish extract composition confirmed that it contained various types of dietary sugars. In particular, dietary sugars with a high uronic acid content exhibit diverse physiological activities [53]; therefore, they are expected to show excellent efficacy.

We analyzed the prebiotic activity of a newly discovered RGP by measuring its effects on media pH and SCFAs concentrations and by assessing cell viability and lipid accumulation levels in 3T3-L1 adipocytes. Our findings suggest that RGP has prebiotic effects, evidenced by changes in SCFAs concentrations and media pH. The gut microbiota plays a crucial role in human health and disease, and its composition can be modulated by dietary factors. High-fat diets promote the growth of potentially harmful bacteria such as Firmicutes in the gut and lead to various metabolic disorders by generating harmful compounds and increasing the gut pH [54]. In contrast, the consumption of high-fiber diets can promote the growth of beneficial bacteria that ferment indigestible polysaccharides to produce SCFAs and lower gut pH, leading to improved gut barrier function and reduced inflammation [2]. Our findings demonstrated that probiotic strains effectively utilized RGP, suggesting its potential as a prebiotic.

The observed decrease in pH indicated organic acid generation, including phenolic and short-chain fatty acids, which are responsible for various aspects of bodily health [55]. Our analysis corroborated that this observed decrease resulted from consuming some extracts by *Lactobacillus* and *Bifidobacterium* [56].

The gut microbiota not only plays a crucial role in the breakdown and absorption of complex carbohydrates and fibers, but also produces SCFAs such as acetate, propionate, and butyrate in the intestine. SCFAs are the main byproducts of microbial fermentation of carbohydrates and play a role in suppressing pathogenic intestinal bacteria, regulating lipid metabolism, and modulating the immune system [57]. The major SCFAs in the gut, including butyrate, propionate, and acetate, are found in a ratio of approximately 60:20:20 in the colon and feces [58,59], and they contribute to the beneficial effects of the gut microbiota by lowering the pH of the gut environment and promoting the growth of beneficial bacteria. These fatty acid metabolites are involved in various important functions, including modulating the immune system, vitamin synthesis, and protection of the intestinal mucosa against pathogens [60]. Furthermore, numerous studies have been conducted on the effects of probiotics on SCFAs production in the human gut microbiota [61].

Probiotics are live microorganisms that, when administered in adequate amounts, confer health benefits to the host by maintaining the balance of gut microbiota and improving gut function. To date, research on probiotics has focused mainly on lactic acid bacteria, particularly strains of *Lactobacillus* and *Bifidobacterium*. To investigate the effects of probiotic consumption according to age, *L. plantarum* P-8 was administered orally, leading to a significant increase in the concentrations of acetate and propionate [62]. Similarly, we confirmed the prebiotic effects of RGP and found a significant increase in the prebiotic activity of all five strains compared to inulin.

Additionally, we observed that RGP inhibited fat accumulation in 3T3-L1 adipocytes. The preadipose cell line 3T3-L1 is clonally expanded and originally derived from murine Swiss 3T3 cells [63]. In line with our findings, the inhibitory effect on adipogenesis in 3T3-L1 preadipocytes has also been observed with Fucoidan, a polysaccharide derived from seaweed [64]. Scientific evidence has demonstrated the effectiveness of probiotics in regulating adipocytes in both humans [65] and animals [66]. Our findings indicated that even with an increased concentration of RGP during cultivation, cell viability did not decrease, indicating low toxicity. In addition, a significant decrease in lipid accumulation was observed, suggesting that RGP may effectively reduce obesity.

Our study provides evidence for the potential of the RGP to exhibit prebiotic effects and inhibit fat accumulation, thus reducing obesity. However, to further substantiate this potential, it is necessary to investigate the genetic and protein changes associated with lipid accumulation inhibition. Unfortunately, due to limitations in sample quantity, we were unable to conduct studies on gene and protein changes. As part of future research, we plan to address this limitation by conducting studies on gene and protein changes in an in vivo study, thereby complementing our findings. Another limitation of this study is that each phenolic and polysaccharide component present in RGP was not evaluated separately. In natural sources, phenolic compounds have been found to exist in a bound and/or free form with polysaccharides [67]. In future studies, we purify an RGP to verify the efficacy of the unique polysaccharide.

Another limitation of our study is the insufficient investigation of the polysaccharide fraction from radish greens in relation to other microbes present in the human gastrointestinal tract, e.g., *Helicobacter pylori* or *Helicobacter felis*. This falls short of providing a complete definition of prebiotics. However, our findings demonstrate the prebiotic effects of the radish green polysaccharide extract in promoting gut health. Further investigations to assess its effects on gastric health would provide a more comprehensive understanding of its potential as a complete prebiotic. Also, further research is needed to elucidate the structural aspects of RGP and its underlying functional mechanisms in vivo.

Several experimental studies have shed light on the underlying mechanisms through which plant extracts exert beneficial effects on obesity and related conditions, revealing a strong correlation with modifications in the gut microbiota. However, several questions remain unanswered. These include determining the efficacy of plant extracts from different conditions, understanding the specific mechanisms by which plant extracts influence gut microbiota, and investigating the anti-adipogenic effects of plant extracts in diverse animal models and human subjects [35].

In conclusion, this study demonstrated that RGP had prebiotic and anti-adipogenic effects in vitro. The prebiotic effects of RGP were observed on probiotic strains, specifically *Lacticaseibacillus*, *Lactobacillus*, and *Bifidobacterium*, as evidenced by increased prebiotic scores and concentrations of total SCFAs. RGP could modulate the bacterial growth of beneficial gut microbes and inhibit the differentiation and lipid accumulation of adipocytes. These findings suggest that RGP may have potential applications as a functional food ingredients for preventing or treating obesity and related metabolic disorders. Furthermore, this study has demonstrated the potential use of polysaccharide extracts from easily accessible food sources as functional ingredients for promoting gut health. Moreover, our research confirms its suitability for the development of health-oriented materials and supports further endeavors in this direction. These findings suggest that RGP, as a novel polysaccharide, holds promise as a natural functional ingredient in the food industry.

## Figures and Tables

**Figure 1 microorganisms-11-01862-f001:**
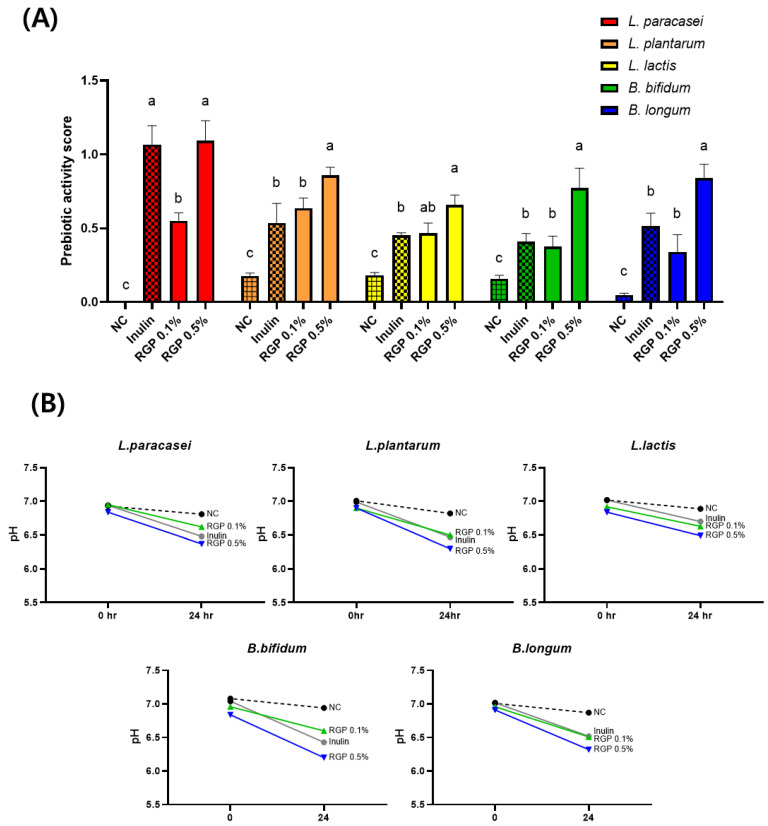
Prebiotic effect of radish green polysaccharide (RGP) on five probiotic strains for 24 h co-culture. (**A**) prebiotic activity score and (**B**) changes in pH. The different letters indicate significant difference (*p* < 0.05) determined by Duncan’s multiple range test.

**Figure 2 microorganisms-11-01862-f002:**
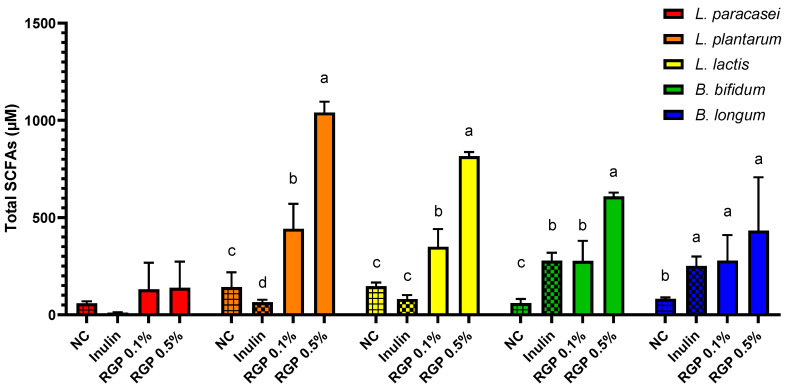
The concentration of total short-chain fatty acids produced by five probiotic strains cultured with radish green polysaccharide (RGP). The different letters indicate significant difference (*p* < 0.05) determined by Duncan’s multiple range test.

**Figure 3 microorganisms-11-01862-f003:**
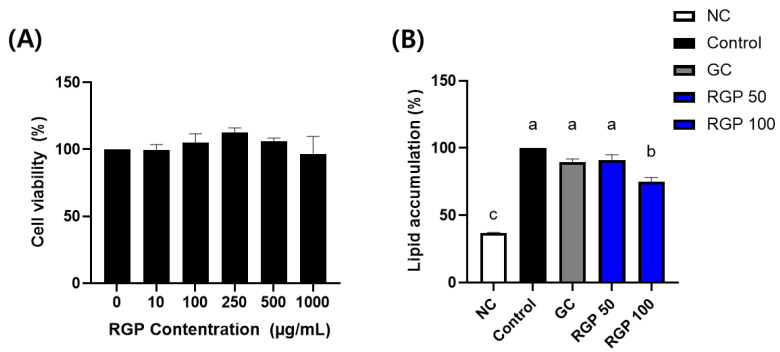
Inhibitory effect of radish greens polysaccharide (RGP) on lipid accumulation in 3T3-L1 cells. (**A**) Cell viability and (**B**) oil-red O-stained area. GC, *Garcinia gummi-gutta* as a positive control. The different letters indicate significant difference (*p* < 0.05) determined by Duncan’s multiple range test.

**Table 1 microorganisms-11-01862-t001:** Chemical properties of radish greens polysaccharide (RGP).

Chemical Property (% of Dry Matter)	RGP
Neutral sugar	68.1 ± 1.3
Uronic acid	20.4 ± 1.5
Protein	1.4 ± 0.3
KDO-like material ^1^	2.2 ± 0.2
Polyphenol	6.5 ± 0.4
**Monosaccharides Composition (Mole %)**	
Rhamnose	6.0 ± 0.4
Fucose	4.8 ± 0.6
Arabinose	24.2 ± 1.8
Xylose	0.4 ± 0.1
Mannose	2.2 ± 0.4
Galactose	42.1 ± 3.0
Glucose	4.2 ± 0.5
Glucuronic acid	2.1 ± 0.3
Galacturonic acid	14.0 ± 2.1

^1^ KDO means 2-keto-3-deoxy-D-manno-octulosonic acid.

## Data Availability

The data presented in this study are openly available. Data available in a publicly accessible repository.

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
