# Peer review of "Prebiotic and Anti-Adipogenic Effects of Radish Green Polysaccharide"

_microorganisms, 2023, doi:10.3390/microorganisms11071862_

Round 1
Reviewer 1 Report
This manuscript describes the results of a study on the prebiotic and anti-adipogenic effects of the polysaccharide fraction from radish greens. The research conducted is extremely important in terms of food production and its effects on the human body, including microorganisms used in food or found in the human digestive system. The research is carried out very imaginatively and the analysis techniques are correctly selected. However, in my opinion the manuscript is fit for publication as it has minor flaws which I list below:
· Lactobacillus paracasei is now classified as Lacticaseibacillus paracasei (see: http://lactobacillus.ualberta.ca/). Please take this into account throughout the text.
· Section „2.7. Statistical analysis” – Please indicate how often the experiments and individual analyzes were repeated.
· The study did not verify the use of the polysaccharide fraction from radish greens by other microorganisms present in the human gastrointestinal tract. As such, it is not known whether these substances meet the full definition of a prebiotic, particularly in terms of selective fermentation by beneficial bacteria. In my opinion, this issue should be discussed when discussing the results.
In my opinion, the manuscript needs minor linguistic corrections (comma, spelling mistakes).
Author Response
We highly appreciate the reviewer’s constructive and helpful comments on our manuscript. As suggested by the reviewer, we have carefully response (marked in blue) to address the reviewer’s comments and revised manuscript (marked in red). We hope that the reviewer will find our responses to the comments satisfactory.
Reviewer #1:
This manuscript describes the results of a study on the prebiotic and anti-adipogenic effects of the polysaccharide fraction from radish greens. The research conducted is extremely important in terms of food production and its effects on the human body, including microorganisms used in food or found in the human digestive system. The research is carried out very imaginatively and the analysis techniques are correctly selected. However, in my opinion the manuscript is fit for publication as it has minor flaws which I list below:
- Lactobacillus paracasei is now classified as Lacticaseibacillus paracasei (see: http://lactobacillus.ualberta.ca/). Please take this into account throughout the text.
â–¶ According to the reviewer’s comment, we revised the words to "Lactocaseibacillus paracasei".
- Section „2.7. Statistical analysis” – Please indicate how often the experiments and individual analyzes were repeated.
â–¶ According to the reviewer’s comment, we added sentence “All experiments were conducted using a triple repetition experimental design.” in Section 2.7.
Lines 163-164: All experiments were conducted using a triple repetition experimental design.
- The study did not verify the use of the polysaccharide fraction from radish greens by other microorganisms present in the human gastrointestinal tract. As such, it is not known whether these substances meet the full definition of a prebiotic, particularly in terms of selective fermentation by beneficial bacteria. In my opinion, this issue should be discussed when discussing the results.
â–¶ Including a larger variety of bacteria from the gastrointestinal tract in experiments is important to confirm the full effectiveness of prebiotics. This aspect was added as a limitation of our study in the Discussion section, highlighting the need for further investigation to achieve a comprehensive understanding of the effects of prebiotics.
Lines 321-328: Another limitation of our study is the insufficient investigation of the polysaccharide fraction from radish greens in relation to other microbes present in the human gastrointestinal tract, e.g. Helicobacter pylori or Helicobacter felis. This falls short of providing a complete definition of prebiotics. However, our findings demonstrate the prebiotic effects of the radish green polysaccharide extract in promoting gut health. Further investigations to assess its effects on gastric health would provide a more comprehensive understanding of its potential as a complete prebiotic. Also, further research is needed to elucidate the structural aspects of RGP and its underlying functional mechanisms in vivo.
※ We would like to thank the reviewers and editor for the constructive and insightful comments and hope that our answers will be acceptable to the reviewer.

Reviewer 2 Report
This experimental study investigates the prebiotic and anti-adipogenic effects of radish greens polysaccharide.
The introduction is nicely written, describing various health benefits such as anti-cancer, anti-diabetic, anti-hypertensive, and anti-obesity effects of Radish (Raphanus sativus) that belong to the Brassicaceae family. Additionally, the authors explained the importance of gastrointestinal microbiota in overall health and the role of dietary fibers and their metabolites in maintaining intestinal homeostasis.
The methodology section provides all necessary information regarding the preparation of radish greens polysaccharide (RGP), composition analysis of RGP, prebiotic activity assay, SCFAs analysis, and in vitro assays.
Results are clearly presented in Table and Figures. In this part, the authors should pay attention to technical elements (e.g., incompleted sentence in line 176, mistakes in lines 113 and 194, etc.).
In the discussion, the authors should emphasize the novelty of the study.
The paper complies with the field of this journal.
Minor editing of English language required
Author Response
We highly appreciate the reviewer’s constructive and helpful comments on our manuscript. As suggested by the reviewer, we have carefully response (marked in blue) to address the reviewer’s comments and revised manuscript (marked in red). We hope that the reviewer will find our responses to the comments satisfactory.
Reviewer #2:
This experimental study investigates the prebiotic and anti-adipogenic effects of radish greens polysaccharide.
The introduction is nicely written, describing various health benefits such as anti-cancer, anti-diabetic, anti-hypertensive, and anti-obesity effects of Radish (Raphanus sativus) that belong to the Brassicaceae family. Additionally, the authors explained the importance of gastrointestinal microbiota in overall health and the role of dietary fibers and their metabolites in maintaining intestinal homeostasis.
The methodology section provides all necessary information regarding the preparation of radish greens polysaccharide (RGP), composition analysis of RGP, prebiotic activity assay, SCFAs analysis, and in vitro assays.
â–¶ We appreciated reviewer’s kind comment.
Results are clearly presented in Table and Figures. In this part, the authors should pay attention to technical elements (e.g., incompleted sentence in line 176, mistakes in lines 113 and 194, etc.).
â–¶ In response to the reviewer's comments, we delete the incompleted sentence, and revised the comma and spelling errors in our manuscript.
Line 176: The prebiotic activity score evaluation for all bacteria revealed that 0.5% RGP showed the highest activity compared to the positive control inulin.
Line 123: broth.
Line 201: strains
In the discussion, the authors should emphasize the novelty of the study.
â–¶ In response to the reviewer's comment, we have included additional sentences regarding the novelty of our study in Discussion section. We conducted research on functional ingredients that can enhance gut health using easily accessible food sources. This highlights the potential of natural functional substances to promote overall well-being.
Lines 336-347: In conclusion, this study demonstrated that RGP had prebiotic and anti-adipogenic effects in vitro. The prebiotic effects of RGP were observed on probiotic strains, specifically Lacticaseibacillus, Lactobacillus and Bifidobacterium, as evidenced by increased prebiotic scores and concentrations of total SCFAs. RGP could modulate bacterial growth of beneficial gut microbes and inhibit the differentiation and lipid accumulation of adipocytes. These findings suggest that RGP may have potential applications as a functional food ingredient for preventing or treating obesity and related metabolic disorders. Furthermore, this study has demonstrated the potential use of polysaccharide extracts from easily accessible food sources as functional ingredients for promoting gut health. Moreover, our research confirms its suitability for the development of health-oriented materials and supports further endeavors in this direction. These findings suggest that RGP, as a novel polysaccharide, holds promise as a natural functional ingredient in the food industry.
The paper complies with the field of this journal.
â–¶ We appreciated reviewer’s comment.
※ We would like to thank the reviewers and editor for the constructive and insightful comments and hope that our answers will be acceptable to the reviewer.

Round 2
Reviewer 1 Report
I wanted to reach out to express my sincere appreciation for the Authors you made to the manuscript. Thanks to your efforts, the quality of work has improved significantly and I am satisfied with the current version. I accept the current version of the manuscript for further processing by the editorial board of the journal.
Author Response
We highly appreciate the reviewer’s constructive and helpful comments on our manuscript. As suggested by the reviewer, we have carefully response (marked in blue) to address the reviewer’s comments and revised manuscript (marked in red). We hope that the reviewer will find our responses to the comments satisfactory.
Reviewer 1:
I wanted to reach out to express my sincere appreciation for the Authors you made to the manuscript. Thanks to your efforts, the quality of work has improved significantly and I am satisfied with the current version. I accept the current version of the manuscript for further processing by the editorial board of the journal.
â–¶ We are glad you are satisfied with our response. I am sure that the reviewer's comments have improved the manuscript.
Academic editor:
- Table 1 re chemical property % and monosacch moles % are the values means (SD)? or something else?
â–¶ We appreciate to editor’s kind comment. Ambiguous texts have been corrected.
Lines 172-173: All data are presented as mean ± standard deviation (n = 3).
Line 174: Table 1. Chemical properties of radish greens polysaccharide (RGP)
|
Chemical property (% of dry matter) |
RGP |
|
Neutral sugar |
68.1 ± 1.3 |
|
Uronic acid |
20.4 ± 1.5 |
|
Protein |
1.4 ± 0.3 |
|
KDO-like material1 |
2.2 ± 0.2 |
|
Polyphenol |
6.5 ± 0.4 |
|
Monosaccharides composition (Mole %) |
|
|
Rhamnose |
6.0 ± 0.4 |
|
Fucose |
4.8 ± 0.6 |
|
Arabinose |
24.2 ± 1.8 |
|
Xylose |
0.4 ± 0.1 |
|
Mannose |
2.2 ± 0.4 |
|
Galactose |
42.1 ± 3.0 |
|
Glucose |
4.2 ± 0.5 |
|
Glucuronic acid |
2.1 ± 0.3 |
|
Galacturonic acid |
14.0 ± 2.1 |
1 KDO means 2-keto-3-deoxy-D-manno-octulosonic acid.
- Table 1 is it PRG or RGP as in the text?
â–¶ Table 1 is incorrectly written by us, and is correct it as follows;
Line 174: Table 1. Chemical properties of radish greens polysaccharide (RGP)
|
Chemical property (% of dry matter) |
RGP |
|
Neutral sugar |
68.1 ± 1.3 |
|
Uronic acid |
20.4 ± 1.5 |
|
Protein |
1.4 ± 0.3 |
|
KDO-like material1 |
2.2 ± 0.2 |
|
Polyphenol |
6.5 ± 0.4 |
|
Monosaccharides composition (Mole %) |
|
|
Rhamnose |
6.0 ± 0.4 |
|
Fucose |
4.8 ± 0.6 |
|
Arabinose |
24.2 ± 1.8 |
|
Xylose |
0.4 ± 0.1 |
|
Mannose |
2.2 ± 0.4 |
|
Galactose |
42.1 ± 3.0 |
|
Glucose |
4.2 ± 0.5 |
|
Glucuronic acid |
2.1 ± 0.3 |
|
Galacturonic acid |
14.0 ± 2.1 |
1 KDO means 2-keto-3-deoxy-D-manno-octulosonic acid.
※ We would like to thank the reviewers and editor for the constructive and insightful comments and hope that our answers will be acceptable to the reviewer.
